# OpenReview forum: "Diverse Preference Learning for Capabilities and Alignment"
_ICLR.cc/2025/Conference — ICLR 2025 Poster_

### Official Review · Reviewer_whFb · 2024-10-25

**Soundness:** 2
**Presentation:** 2
**Contribution:** 2
**Rating:** 5
**Confidence:** 4

**Summary:**

The paper discusses how alignment algorithms like RLHF and DPO reduce the diversity of large language models (LLMs), leading to less varied and overly homogeneous outputs. The study identifies the KL divergence regularizer in preference learning algorithms as a primary cause, which biases models towards majority opinions at the expense of diversity. To combat this, the paper proposes a new method, Diverse Preference Learning (DPL), which decouples entropy and cross-entropy terms in the KL penalty to allow more controlled diversity in LLM outputs. DPL demonstrates better performance in representing a broader range of societal perspectives and produces outputs with greater semantic and lexical diversity compared to traditional methods.

**Strengths:**

1. The theoretical analysis on KL-divergence sounds interesting.
2. The experimental design of the article is quite creative.

**Weaknesses:**

1. The objective of DPL (Eq. 9) appears interesting, yet the derivation process seems to have many inconsistencies, as detailed in the Questions section.
2. You should define Proposition 3.1, Corollary 3.2, and Proposition A for clarity.
3. The experiments lack comparisons with existing Diverse Preference Learning methods, such as [1,2], and are missing richer alignment baselines for analysis.
4. The experiments lack many implementation details necessary for readers to fully understand or reproduce the results. See Questions for details.
5. It appears that DPL does not perform better than DPO on mathematical tasks.

[1] Beyond Reverse KL: Generalizing Direct Preference Optimization with Diverse Divergence Constraints
[2] Inverse-RLignment: Inverse Reinforcement Learning from Demonstrations for LLM Alignment

**Questions:**

1. In line 215, what if ``Empirical choices of β typically lie in the range [0.01, 0.1].''
2. Question towards Sec. 3.2:
  2.1. In Eq. 8, should it be H(π(·|x),πref (·|x))?
  2.2. In Eq. 22, should it be + απ^⊤ log π?
  2.3. In Eq. 23, should it be + α log π?
  2.4. From Eq. 25 to Eq. 26, should π = exp()-1?
  2.5. If Question 5 stands, does Eq. 26 stand?
  2.6. If in Eq. 28, it is proportional relation, you cannot use equal in Eq. 29.
3. How to calculate "1) general semantic diversity, 2) logical diversity or diversity of viewpoints, and 3) content diversity." and what's their meaning? What is "Embedding Cosine Distanc" in Figure 2?
4. Why do you generate 16 responses for each question?
5. How do you train the reward model, and why the rewards are all negative?
6. What is DPO t=1.2 in Figure3, and where is the definition of t?
7. Why do you use Mistral-7B-Instruct-v0.2 for DIVERSITY-QUALITY TRADEOFFS, while Mistral-7B base for the other two task?

---

> ### Author Response · Authors · 2024-11-21
>
> We thank the reviewer for their thorough technical feedback. We address each point below:
>
> **Theoretical Analysis and Derivations:**
> * We have corrected the copying errors in Equations 8, 25, and 26 in the appendix. These were transcription mistakes and do not affect the theoretical results. We thank the reviewer for catching these.
> * Equations 22 and 23 are correct as written, since entropy has a negative sign: $H(\pi) \= -\sum_y \pi(y) \log \pi(y)$
> * We've revised notation to use proportional symbols consistently
> * We've renamed propositions and corollaries to reflect their section locations for clarity
>
> **Implementation Details:** We have expanded Appendix B with comprehensive experimental details (including for all training runs) to aid reproducibility. We will also release our code upon acceptance. We include a few key pieces of information here:
>
> * **Reward model:** Trained using Mistral-7B-Instruct with LoRA and reward head on HH-RLHF using Bradley-Terry loss. We note that since the Bradley-Terry loss is shift-invariant, only relative differences between reward values are meaningful.
> * **Diversity metrics:** We updated the Section 4.1. Diversity Metrics section to connect our concrete diversity metrics with the conceptual types of diversity they measure: “our diversity metrics roughly measure 1\) general semantic diversity (embedding distance metric), 2\) logical diversity or diversity of viewpoints (logical disagreement metric), and 3\) diversity in response content and ideas (content diversity metric)”. We also updated the appendix with the mathematical formula for embedding cosine distance and details on the embedding model used
> * **Sample generation:** We use 16 samples per prompt to obtain robust per-input diversity measurements and low-variance quality estimates
> * **Hyperparameters:** Hyperparameters and training data specifications for all training runs are now included
>
> **Choice of Models:** We used Mistral-7B-Instruct for diversity experiments as it performed well in chat settings. For mathematical reasoning and calibration, we found better results using the Zephyr finetuning approach \[1\] (which starts from Mistral-7B base), as it is a stronger finetuning recipe than using HH-RLHF. For consistency, we will update all experiments in the main paper to use the Mistral-base/Zephyr setup in the camera-ready version.
>
> **Additional Baselines:** Thank you for highlighting relevant work on diverse preference learning. We have already implemented and will include full results comparing against f-DPO in the camera-ready version. However, Inverse RLignment addresses a fundamentally different setting (learning from demonstrations) than our work, which focuses specifically on understanding and addressing diversity loss in pairwise preference learning algorithms such as RLHF and DPO. Our theoretical analysis of KL regularization and proposed solution are specific to the preference learning setting, making direct empirical comparisons less meaningful. We have added a discussion of Inverse RLignment to our related work section to highlight these complementary approaches.
>
> **Mathematical Task Performance:** While DPL shows mixed results on easy/medium difficulty problems, it demonstrates consistent improvements on hard problems in terms of accuracy and sample efficiency:
>
> * GSM8K:
>   * \+10% vs vanilla DPO, \+4% vs temperature-scaled DPO at 128 samples
>   * DPL needs 84 samples to match DPO@128 (34% savings)
>   * DPL needs 106 samples to match temp-scaled DPO@128 (17% savings)
> * MATH:
>   * \+4% vs vanilla DPO, \+1% vs temperature-scaled DPO at 128 samples
>   * DPL needs 88 samples to match DPO@128 (31% savings)
>   * DPL needs 113 samples to match temp-scaled DPO@128 (12% savings)
>
> Regarding the meaning of DPO $t=1.2$ in Figure 3 – throughout the paper, we used the notation $t$ to refer to token-level temperature and $\\alpha / \\beta$ to refer to DPL’s global temperature. This is defined mathematically in Section 3.2. However, for the sake of clarity, we updated the Figure 3 caption to explain that t refers to token-level temperature.
>
> **Empirical $\\beta$ Range:** We've strengthened our claim about typical $\\beta$ values with citations to foundational RLHF and DPO works which use $\\beta \\in \[0.01, 0.1\]$: 1\) OpenAI's InstructGPT, 2\) Cohere's REINFORCE paper, 3\) Original DPO paper, 4\) Allen AI's Zephyr technical report
>
> \[1\] [Zephyr: Direct Distillation of LM Alignment](https://arxiv.org/abs/2310.16944)

---

> > ### Comment · Reviewer_whFb · 2024-11-22
> >
> > Thank you for your response; it has addressed most of my concerns. However, I still have reservations about the adequacy and fairness of the performance comparison under the BoN setting with 16 samples per prompt, as it is not a conventional setup. While I understand that DPL can enhance the diversity of responses, I am concerned about potential instability in its performance during regular usage (1 sample per prompt). Therefore, I would appreciate it if the authors could provide empirical explanations, such as the standard deviation or a comparison under the vanilla greedy decoding setting.

---

> > > ### Author Response · Authors · 2024-11-22
> > >
> > > We thank the reviewer for their engagement with our reply. We want to clarify an apparent misunderstanding of our evaluation setup. For the diversity-quality experiments (Section 4.1), we generate 16 responses per prompt but take the **average** reward across all 16 responses, not the maximum. This evaluates expected single-sample performance in chat settings and reflects conventional setups. We generate multiple responses per prompt solely to obtain lower-variance estimates of expected performance.
> > >
> > > We have updated the Section 4.1 Quality Metrics section to make this explicit, writing: "We then evaluate models by averaging their generation rewards across both a held-out test set of 500 inputs and 16 responses per input." We only use a best-of-N setup for mathematical reasoning (Section 4.2), where inference-time scaling methods have become standard practice for challenging reasoning tasks.

---

> > > > ### Comment · Reviewer_whFb · 2024-11-23
> > > >
> > > > Thank you for your response. I have increased my score.

---

> > > > > ### Author Response · Authors · 2024-11-29
> > > > > **New f-DPO Baseline Results**
> > > > >
> > > > > We sincerely thank the reviewer for their updated assessment and continued engagement.
> > > > >
> > > > > We’ve now added comprehensive f-DPO baseline results to the diversity-quality plot (see Figure 5 of the Appendix), addressing one of the reviewer’s earlier requests for an f-DPO baseline comparison. Specifically, we consider $\alpha$-divergences including $\alpha=$ 0 (reverse KL or standard DPO), 0.001, 0.1, 0.3, 0.5, 0.9, and 1 (forward KL).
> > > > >
> > > > > DPL is competitive or dominates f-DPO across all metrics. Furthermore, DPL spans the full range of diversity better than f-DPO, which degrades into unintelligible text beyond $\alpha=1$, which occurs at only the midpoint of our diversity plots.
> > > > >
> > > > > Given this additional baseline and our previous responses addressing your concerns about experimental details and evaluation setup, would you consider raising their score further? We appreciate your feedback and suggestions, which have helped strengthen the paper.

---

### Official Review · Reviewer_foMz · 2024-10-28

**Soundness:** 3
**Presentation:** 4
**Contribution:** 3
**Rating:** 8
**Confidence:** 3

**Summary:**

The contribution of this paper is simple, but well-motivated and addressing a real issue.

The authors note that the $\beta$ parameter, which controls the KL regularisation strength in DPO, is involved in mode collapse to the majority option. Temperature sampling, they argue, does not fix this, as it causes disfluencies and broken outputs. To address this, they propose to decompose KL into an entropy and a cross-entropy term with the former, each with separate weights modulating the strength of the regularisation. They are able to bake this into DPO, proposing a new objective called DPL. They show how this objective can be thought of as temperature scaling on the sequence level.

They perform two different flavours of evaluation: in the first one they assess the quality/diversity tradeoff on Arena-Hard and HH-RLHF and they found DPL to be Pareto-dominant; in the second, they show that in a best-of-N sampling setup (GSM8K and MATH), DPL helps on hard problems, but results are mixed on easy-medium. Finally, they show improved results on MMLU and TruthfulQA in terms of both accuracy and calibration error.

**Strengths:**

- To the best of my knowledge, the proposed method, while obvious in hindsight (as most good things are) is novel
- The problem is well posed and mostly convincingly discussed. Acting on the entropy of the distribution has been gaining momentum in the community
- Experiments are quite comprehensive, though some baselines should probably be included. (This is the main reason I am giving a 6 instead of an 8) (edit: improved after rebuttal)
- The quality of the writing is outstanding

**Weaknesses:**

- The experimental setting could be stronger: DPL is mostly pitted against vanilla temperature scaling, which on its own does lead to bad outputs. However, top-k, top-p, and, more recently, min-p sampling are all relatively established solution that should have been incorporated in the analysis. (edit: improved after rebuttal)
- Basic REINFORCE (with no KLD in the reward), which has become very popular again without should also have been compared against. This is more of a desideratum than a hard requirement.
- Figure 2 seems to show some brittleness of DPL, with many green dots substantially worse than the Pareto frontier. Some additional discussion about it would be helpful

**Questions:**

Suggestions:
- Figure 3 is hard to read

---

> ### Author Response · Authors · 2024-11-21
> **Adding Baselines and Addressing Figures**
>
> We sincerely thank the reviewer for their positive and constructive feedback. We are glad they found our analysis of the mode collapse problem well-posed and convincing. We are also glad they appreciated our principled solution and found our experiments comprehensive.
>
> We address their comments and questions below:
>
> **Missing Baselines:** We thank the reviewer for the suggestion that including more thorough baselines would provide additional context for evaluating our method's performance. We have already implemented and will include full results comparing against top-k, top-p, and min-p sampling in the camera-ready version. We have updated Figure 2 with complete min-p experiments and will add additional results as they finish.
>
> Preliminary results show DPL outperforms min-p sampling on the embedding and logical disagreement metrics, especially at high temperatures. However, on the content diversity metric, min-p performs slightly better than DPL. This is related to the non-monotonicity of this metric at higher temperatures. This is where DPL shines, and thus DPL looks worse on this metric. We discuss this in the next section of this review.
>
> Regarding REINFORCE, we note that recent work replacing PPO with REINFORCE for LLM alignment (e.g., Wei et al. 2024\) still incorporates KL regularization in the RL objective. However, if the reviewer had a specific REINFORCE variant in mind that excludes KL terms, we would be very interested in potentially including it as an additional baseline.
>
> **Points Below the Pareto Frontier (Figure 2):** The reviewer raises a good point about the apparent brittleness suggested by points far below the Pareto frontier. These points reflect a phenomenon where high temperatures can actually decrease measured diversity on certain prompt-based metrics. This causes them to appear below the Pareto curve instead of to the right of it. Specifically:
>
> * Content, surface form, and perspective diversity metrics (Appendix B) show non-monotonic relationships with high temperatures.
> * Embedding-based and logical diversity metrics increase monotonically
> * We clipped extreme outliers that would have appeared far to the left of the vanilla DPO point (these points were always token-level scaled or token-level scaled with min\_p, never DPL). This hides some of the non-monotonic behavior for these methods.
>
> We updated the results section of Section 4.1 with this discussion. We also labeled points exhibiting this kind of non-monotonic behavior with their parameter values in Figure 2 and Appendix B.
>
> **Figure 3 Readability:** We thank both reviewers for noting the difficulty in reading Figure 3\. We have updated it with:
>
> * An improved color scheme (switching from green/blue to orange/blue)
> * Greater contrast between different temperature gradients
> * Added best-of-N accuracy numbers in tabular form in Appendix C
>
> We thank the reviewer for their comments again and will continue to update the paper as we obtain full baseline results with top-p and top-k sampling.

---

> ### Author Response · Authors · 2024-11-28
> **Additional Baseline Comparisons and Paper Updates**
>
> We thank the reviewer for their constructive feedback, particularly their suggestion to include additional baselines, which has helped strengthen our empirical validation.
>
> We're pleased to share that we have now included comprehensive baseline comparisons in Figure 2, including:
>
> 1. Min-p sampling
> 2. Top-p (nucleus) sampling
> 3. Top-k sampling
> 4. [f-DPO](https://arxiv.org/abs/2309.16240), another preference learning algorithm intended to preserve output diversity (these results are in Figure 5 of the Appendix due to space constraints)
>
> These results show DPL maintains competitive or superior performance against these established methods, particularly on embedding distance and logical disagreement metrics.
>
> We have also enhanced the paper with:
>
> * Improved Figure 3 readability with a new column showing accuracy gains relative to DPO
> * An extensive theoretical and empirical analysis in Appendix C examining the relationship between temperature, diversity, and problem-solving performance
>
> Given that we've addressed the reviewer’s concerns about baseline comparisons and Figure 2, would the reviewer consider raising their score? We thank the reviewer again for their feedback and would be happy to address any remaining questions or concerns.

---

> > ### Comment · Reviewer_foMz · 2024-12-01
> > **Rebuttal acknowledgement**
> >
> > Thanks for the response. I have increased my score.

---

> > > ### Author Response · Authors · 2024-12-02
> > >
> > > We sincerely thank the reviewer for their engagement and updated assessment.

---

### Official Review · Reviewer_Tm8X · 2024-10-30

**Soundness:** 4
**Presentation:** 4
**Contribution:** 3
**Rating:** 8
**Confidence:** 4

**Summary:**

This paper addresses the issue that alignment algorithms lead to a reduction in diversity. They motivate the need for diversity in a few ways: 1) societal implications, 2) better diversity might mean better inference-time problem solving (because you’re essentially doing a wider search in solution space), 3) better calibration.

Though temperature scaling is one way of addressing this issue, the authors demonstrate that it is not sufficient, as temperature scaling quickly leads to degradation in generation quality.

The authors address the problem by first observing that the KL divergence term is the culprit for drop in diversity (Proposition 1). Ie, the hyperparameter \beta controls both the reference policy’s weighting and the sharpness of the output distribution.
The solution that the authors propose is to decouple the KL regularization term into a cross-entropy term and entropy term, but weigh them independently (with hyperparameters \alpha, \beta), which in turn ends up independently affecting the reference model’s output distribution and how much such distribution is weighted (proposition 2).

The authors draw a couple of connections between their change and prior work – when \alpha == \beta, then this is the standard DPO regime.
Similarly, their \alpha parameter can be thought of as temperature scaling, but done at a sequence level as opposed to token level (equation 14).

Finally, the authors demonstrate a series of empirical results that match their original set of motivations (1: better diversity, 2: better inference-time problem solving, 3: better calibration).

1) Diversity (Figure 2): The suggested approach method achieves better pareto curves when plotting diversity scores vs. generation quality
2) Problem solving (Figure 3): For hard problems, in which an increased number of generations helps solve the task, their suggested approach leads to higher performance, which they attribute to increase in diversity across generations (ie, doing a wider search in solution space).
3) Calibration (Figure 4): The authors show that *without affecting accuracy*, their suggested approach leads to improved calibration (according to ECE and Brier scores).

**Strengths:**

Well organized, lots of empirical results, the suggested approach is simple and principled.

**Weaknesses:**

* At a high level, the authors attribute improved problem solving (which is actually not very evident in Figure 3) to improved diversity. Though I could imagine that diversity hypothetically can lead to improved problem-solving performance for methods that use inference-time scaling (because the model is essentially generating more hypotheses for solving a problem), it seems like Figure 3 is not quite telling this story.
* First of all, in the case of easy/medium difficulty problems, their suggested approach does not seem to be better than DPO – this is fine for the easy case – in which both DPO and DPL saturate to near perfect accuracy, meaning that the problems didn’t require a wide search in hypotheses in the first place.
* However, in the medium case, DPO outperforms DPL, with still a lot of room for improvement for both models (ie. we do not see high accuracy saturation).
* Though DPL outperforms DPO in the hard case, the improvement is rather minimal.

* Perhaps more importantly, if more diversity does indeed lead to better problem solving, then even within DPL, we should see a relationship between temperature and best-of-N accuracy, which is not the case in Figure 3 – sometimes higher temperature is better, sometimes it’s not. Studying how accuracy is affected by varying temperatures would have been helpful.
* The author’s claim that improved diversity leads to better problem solving for methods that use inference-time scaling could be strengthened by actually trying an inference-time scaling approach – this is something that can be empirically studied. I realize this might be a big ask, but I am willing to raise my score significantly if indeed the authors demonstrate evidence of this. Otherwise, I think their claim that DPL leads to better problem-solving is not very convincing.
* (While we’re at it - minor note regarding Figure 3: the color scheme (gradients of green + blue) makes it very very difficult to parse the results. Please consider changing it)

**Questions:**

Figure 2: There seems to be quite a large range in terms of win-rate/avg. reward/ref. policy CE when you consider points that lie below the Pareto curve (points with lighter shades). Do you have an explanation for such behavior? If the range of behavior is so wide, is it possible that the improved pareto curve is due to noise?

---

> ### Author Response · Authors · 2024-11-21
> **Clarifying Diversity-Performance Relationships and Core Contributions**
>
> We sincerely thank the reviewer for their thoughtful and detailed feedback. We appreciate their acknowledgment of our method's principled approach and comprehensive results. Below, we address the reviewer’s main points and questions.
>
> **Relationship Between Diversity and Problem-Solving Performance:** The reviewer raises important questions about the relationship between temperature and problem-solving performance. We have significantly expanded our analysis to better illustrate these relationships:
>
> 1. Quantitative Improvements: While the visual presentation in Figure 3 may have obscured the gains, DPL shows consistent, meaningful improvements over baselines on hard splits:
> * GSM8K:
>   * \+10% vs vanilla DPO, \+4% vs temperature-scaled DPO at 128 samples
>   * DPL needs 84 samples to match DPO@128 (34% savings)
>   * DPL needs 106 samples to match temp-scaled DPO@128 (17% savings)
> * MATH:
>   * \+4% vs vanilla DPO, \+1% vs temperature-scaled DPO at 128 samples
>   * DPL needs 88 samples to match DPO@128 (31% savings)
>   * DPL needs 113 samples to match temp-scaled DPO@128 (12% savings)
> 2. Temperature-Performance Relationship: We have added an extensive analysis and discussion in Appendix C showing that:
>    * Each problem has a unique optimal temperature, which is related to the problem’s difficulty (i.e. single-sample success probability). In particular
>      1. Easy problems benefit from lower temperatures as they don't require extensive exploration
>      2. Medium problems show mixed results as they operate in a transition regime
>      3. Hard problems benefit from higher temperatures up to an optimal point, after which DPL degrades more gracefully than token-level scaling
>    * Best-of-N sampling benefits from higher variance sampling strategies as N increases. We tie this to the concavity of the best-of-N success rate function.
>      Our added analysis provides a theoretical explanation for the observed behavior and empirical evidence supporting this theory.
> 3. Best-of-N as Inference-Time Scaling: We note that best-of-N is a widely-used inference-time scaling approach. While we agree that exploring other scaling strategies (e.g., beam search) would be interesting, recent work shows they can underperform best-of-N in high-sample settings (e.g., [https://arxiv.org/abs/2408.03314](https://arxiv.org/abs/2408.03314) Figure 3 left).  Best-of-N sampling also has the advantage of relying only on additional samples, so improved performance of DPL does not correlate with other aspects of the inference-time scaling method. Overall, we believe best-of-N is a sufficiently robust, popular, and representative inference-time scaling approach to evaluate our method on.
>
> **Figure 3 Readability:** We thank the reviewer for noting the difficulty in reading Figure 3\. We have:
>
> * Changed to a more distinct color scheme (orange/blue)
> * Increased contrast between temperature gradients
> * Added results in tabular form to Appendix C
>
> **Cause of Points Below the Pareto Frontier (Figure 2):** These points reflect a phenomenon where high temperatures can actually decrease measured diversity on certain prompt-based metrics. This causes them to appear below the Pareto curve instead of to the right of it. Specifically:
>
> * Content, surface form, and perspective diversity metrics (Appendix B) show non-monotonic relationships at high temperatures
> * Embedding-based and logical diversity metrics increase monotonically
> * We clipped extreme outliers that would have appeared far to the left of the vanilla DPO point (these points were always token-level scaled or token-level scaled with min\_p, never DPL). This hides some of the non-monotonic behavior for these methods.
>
> We updated Section 4.1 results with this discussion. We also labeled points exhibiting this kind of non-monotonic behavior with their parameter values in Figure 2 and Appendix B.
>
> **Additional Core Contributions:** While the reviewer focused on problem-solving results, our paper makes several other key contributions:
>
> 1. We identify KL regularization as a central cause of diversity loss in aligned LLMs
> 2. We provide a theoretical analysis using social choice theory to show how KL regularization amplifies majority preferences.
> 3. We propose DPL as a principled solution that provably restores distributional representation.
> 4. We empirically validate our method across multiple settings beyond just problem-solving.
>
> We thank the reviewer again for their thoughtful comments and note that this feedback was very helpful in improving the paper.

---

> > ### Comment · Reviewer_Tm8X · 2024-11-22
> >
> > Thank you for the additional responses
> >
> > * Ah, I did not make the connection that best-of-N is also an inference-time-scaled approach, though it's obvious in hindsight that it is -- my mistake. I have re-read Section 4.2 and Figure 3 with this in mind. (For what it's worth, what I had in mind when writing the review was something that involved "search", such as MCTS or tree-of-thought, though this is less needed now that we've clarified that best-of-N is also an inference-time scaled approach).
> >
> > * Figure 8 is very helpful, and I almost wonder if it could be an added column in Figure 3 to 1) better highlight the benefit of diversity as # of samples is increased, and 2) since the curves in column 3 of Figure 3 are all kind of cluttered. (This is a genuine thought, not an ask I'm making from the authors).
> >
> > * I'm still a bit confused as to why DPO outperforms DPL for the medium difficulty tasks -- Does this mean that while DPL generates more "hypotheses", the best hypotheses from DPL are worse than the hypotheses generated by DPO?
> >
> > * For the time being, I've raised my soundness, presentation, and contribution scores.

---

> > > ### Author Response · Authors · 2024-11-26
> > > **Clarifying DPL's Diversity-Quality Tradeoffs and Best-of-N Performance Analysis**
> > >
> > > We sincerely thank the reviewer for their continued engagement and for raising their attribute-level scores.
> > >
> > > **Best-of-N and Search Strategies:** We're glad we could clarify the connection between best-of-N and inference-time scaling. Your suggestion about MCTS and tree-of-thought approaches is interesting \- while we focused on best-of-N as a simple and widely-used scaling method, exploring DPL's interaction with more sophisticated search strategies could be valuable future work.
> > >
> > > **Improved Figure Clarity**: Thank you for the thoughtful suggestion about Figure 8\. We agree it improves clarity and updated Figure 3 with this additional column. This better illustrates how DPL's benefits emerge as the sample budget increases, particularly on hard problems.
> > >
> > > **Analysis of Medium-Difficulty Performance**: Your interpretation of why DPO outperforms DPL on medium-difficulty problems is exactly right.
> > >
> > > 1. **DPL generates more unique solutions:** We've added Figure 10 in Appendix C, showing that 1\) duplicate solutions are a major cause of sample inefficiency at high N (30-40% of DPO samples are duplicates) and 2\) DPL generates more unique solutions (10-20% fewer duplicates than DPO).
> > > 2. **DPL’s average solution quality is lower – although this gap becomes smaller as problems get harder:** This is visible in single-sample (N=1) performance, where DPO's y-intercept in Figure 3 is considerably higher than DPL on easy and medium problems, but only slightly higher on hard problems. Tables 1-3 in the Appendix give these same numbers in tabular form for easier comparison. We updated Appendix C to include an explicit discussion of this. We connect it back to the optimal temperatures result in Figure 6 – which is about single-sample performance, not best-of-N.
> > > 3. **Whether this tradeoff is beneficial depends on the number of samples and problem difficulty:** On medium problems, DPO’s higher-quality individual solutions outweigh DPL’s increased diversity through N=128. On hard problems, where single-sample performance gaps between DPO and DPL are low (e.g., 0.2% on GSM8K), DPL's diversity advantage as N increases becomes more valuable than small differences in individual solution quality.
> > >
> > > We appreciate the reviewer’s thoughtful engagement which has helped us better articulate these insights. Given that we've addressed the reviewer’s main questions and concerns about DPL's performance, would the reviewer consider raising their score? We thank the reviewer again for their thoughtful feedback and note that we would be happy to address any remaining concerns.

---

> > > > ### Comment · Reviewer_Tm8X · 2024-11-26
> > > >
> > > > Thank you for the response. Given the extensive amount of additional results added in the appendix, I have raised my score.

---

> > > > > ### Author Response · Authors · 2024-11-28
> > > > >
> > > > > We sincerely thank the reviewer for their continued engagement in suggesting and responding to new analyses. Their feedback helped us strengthen both the presentation and technical depth of our paper.

---

### Author Response · Authors · 2024-11-21
**Global Response: Strengthening Baselines, Analysis, and Clarity**

**Global Response:** We sincerely thank all reviewers for their thoughtful and detailed feedback.

The reviewers appreciated several aspects of our work, including its principled approach to addressing diversity loss, strong theoretical foundations connecting KL regularization to mode collapse, and comprehensive empirical validation across multiple settings. The reviews also identified opportunities for improvement that have helped us enhance the paper.

We have made substantial updates to address the common concerns:

**Comprehensive Baseline Evaluation​​:**

* We thank the reviewers for suggesting additional baselines. We have implemented top-k, top-p, min-p sampling and f-DPO comparisons, which are currently running
* We updated the paper with preliminary results using min-p sampling. Here DPL outperforms min-p on embedding and logical disagreement metrics, especially at high temperatures
* We will include complete results comparing against all these baselines in the camera-ready version

**Clarified Mathematical Results and Performance:**

* Added extensive empirical and theoretical analysis in Appendix C showing the relationship between temperature, problem difficulty, and best-of-N performance
* Quantified concrete improvements on hard problems:
  * 10% accuracy gain, 34% sample efficiency gain vs DPO@128 on GSM8K
  * 4% accuracy gain, 31% sample efficiency gain vs DPO@128 on MATH

**Improvements to Figures and Writing:**

* Added discussion on points below Pareto frontier in Figure 2, and labeled these points to justify our claims about them in the discussion
* Improved readability of Figure 3 problem-solving results
* Updated the conclusion by discussing specific scenarios where DPL provides the most benefit

**Beyond addressing specific concerns, we emphasize that our work makes several fundamental contributions:**

1. Identifies KL regularization as a key cause of diversity loss in aligned LLMs
2. Provides social choice theory analysis showing how this amplifies majority preferences
3. Proposes DPL as a principled solution with provable guarantees
4. Demonstrates empirical benefits across multiple important settings

We believe these updates strengthen the paper while maintaining its core contributions toward understanding and addressing an important challenge in language model alignment. The added experiments and analysis provide stronger validation of our approach. We will continue to update the paper with additional baseline results as we obtain them.

---

### Author Response · Authors · 2024-11-28
**Global Response Update: Latest Paper Improvements**

We are pleased that reviewers have found our detailed responses and paper improvements valuable, particularly our in-depth analysis of problem-solving performance and clarity updates, as reflected in their updated assessments.

As the paper updating period closes, we want to summarize the final improvements made to our paper based on reviewer feedback.

**Additional Baselines:**

* Added extensive baseline comparisons to the diversity-quality tradeoff plot (Figure 2), including min-p, top-p, top-k sampling, and f-DPO (in Figure 5 in the Appendix), as suggested by **Reviewers foMz** and **whFb**
* These results show DPL achieves competitive or superior performance against these established methods, particularly on embedding distance and logical disagreement metrics

**Enhanced Problem-Solving Analysis and Clarity:**

* Improved Figure 3 readability with a new column showing relative accuracy gains compared to DPO, better highlighting DPL's benefits on difficult problems
* Additional results and analysis in Appendix C providing a deeper understanding of the relationship between temperature, solution diversity, and problem-solving performance:
  * Quantified duplicate solutions as a major source of sample inefficiency (30-40% of DPO solutions are duplicates at N=128)
  * Showed DPL reduces duplicate solutions by 10-20%, improving sample efficiency at high sample counts
  * Connected these empirical findings with our theoretical analysis to explain DPL's performance patterns across varying problem difficulties

These updates strengthen both the empirical validation of our method and our theoretical understanding of how diversity impacts problem-solving performance. We thank the reviewers for their continued engagement, which has helped guide our final improvements to the manuscript. While the window for paper edits is closing, we would be happy to run any further experiments the reviewers think would be valuable for validating our analysis.

---

### Meta-Review · Area_Chair_shuS · 2024-12-18

**Metareview:**

This paper introduces a new preference learning algorithm: DPL (Diverse Preference Learning) to address the reduced diversity of outputs in DPO. They achieve this by decoupling the entropy and cross-entropy terms in the KL divergence, responsible for overweighting of majority preferences and reduced output diversity in the original formulation. The work presents comparisons with DPO and argues that their method can be more useful than temperature sampling due to their handling of social choice.

**Strengths:** The motivation behind the paper is strong, the approach is intuitive and there are many empirical results to verify the central hypotheses in the paper. The work leads to improved inference-time scaling and calibration.

**Weaknesses:** Some of the improvements over DPO seem smaller. No comparison with reinforcement learning algorithms like PPO is provided (DPO is not technically a reinforcement learning method: https://iclr-blogposts.github.io/2024/blog/rlhf-without-rl/). I cannot see how the method might ultimately address the social choice question in the intro / motivation, despite showing improvements in diversity.

**Reasoning for accept (poster):** See strengths above. The reviewers were all in favor of acceptance after the author discussion which clarified their confusion and addressed their objections.

**Additional Comments On Reviewer Discussion:**

Reviewers questions on the details of the method and some empirical results and presentations were successfully clarified by the reviewers. The reviewers raised objections about some intuitive baselines such as f-DPO and REINFORCE, the first of which was addressed by adding more results by the authors. I was surprised not to see any results or comparison with REINFORCE despite a reviewer (fomZ) pointing this out explicitly.  Other questions raised included the fairness of the performance comparison under the Best-of-N setting with 16 samples per prompt, which is different from the standard setup, this was addressed to some extent by the authors but did not result in the reviewer whFb increasing their score.

---

### Decision · Program_Chairs · 2025-01-22

Accept (Poster)